

# MAPE-ViT: multimodal scene understanding with novel wavelet-augmented Vision Transformer

Muhammad Waqas Ahmed[1], Touseef Sadiq[2], Hameedur Rahman[1], Sulaiman Abdullah Alateyah[3], Mohammed Alnusayri[4], Mohammed Alatiyyah[5] and Dina Abdulaziz AlHammadi[6]

[1] Department of Computer Science, Air University, Islamabad, Pakistan
[2] Centre for Artificial Intelligence Research, Department of Information and Communication Technology, University of Agder, Grimstad, Norway
[3] Department of Computer Engineering, College of Computer, Qassim University, Buraydah, Saudi Arabia
[4] Department of Computer Science, College of Computer and Information Sciences, Jouf University, Sakaka, Saudi Arabia
[5] Department of Computer Science, College of Computer Engineering and Sciences, Prince Sattam Bin Abdulaziz University, Al-Kharj, Saudi Arabia
[6] Department of Information Systems, College of Computer and Information Sciences, Princess Nourah bint Abdulrahman University, Riyadh, Saudi Arabia

Corresponding author
Hameedur Rahman,
hameed.rahman@mail.au.edu.pk

## ABSTRACT

This article introduces Multimodal Adaptive Patch Embedding with Vision Transformer (MAPE-ViT), a novel approach for RGB-D scene classification that effectively addresses fundamental challenges of sensor misalignment, depth noise, and object boundary preservation. Our framework integrates maximally stable extremal regions (MSER) with wavelet coefficients to create comprehensive patch embedding that capture both local and global image features. These MSER-guided patches, incorporating original pixels and multi-scale wavelet information, serve as input to a Vision Transformer, which leverages its attention mechanisms to extract high-level semantic features. The feature discrimination capability is further enhanced through optimization using the Gray Wolf algorithm. The processed features then flow into a dual-stream architecture, where an extreme learning machine handles multi-object classification, while conditional random fields (CRF) manage scene-level categorization. Extensive experimental results demonstrate the effectiveness of our approach, showing significant improvements in classification accuracy compared to existing methods. Our system provides a robust solution for RGB-D scene understanding, particularly in challenging conditions where traditional approaches struggle with sensor artifacts and noise.

## INTRODUCTION

In the field of computer vision, scene understanding involves analyzing images or videos to interpret the visual environment. To achieve this, various techniques and algorithms have been developed to recognize and categorize entities within a scene and understand the

spatial relationships between those entities. Scene understanding plays a pivotal role in diverse applications, including autonomous vehicles (*Chen & Huang, 2017*), context-aware augmented reality (*Tahara, Ushiku & Harada, 2020*), surveillance and security (*Calavia et al., 2012*), and agriculture (*Tsouros, Bibi & Sarigiannidis, 2019*). However, despite the advancements in this area, numerous challenges persist, such as variations in scale and viewpoints, high computational demands, and issues related to generalization and overfitting. Specifically, using RGB-D data introduces additional complexities, including limitations of depth sensors, occlusions, data gaps, and misalignments between depth and color images. These challenges make accurate scene representation and analysis a difficult task.

RGB-D scene classification has advanced significantly in recent years, yet most existing methods continue to rely heavily on convolutional neural networks (CNNs) and other conventional paradigms. As demonstrated by *Eitel et al. (2015)* and *Song, Lichtenberg & Xiao (2017)*, *Ahmed et al. (2024)* CNN approaches are effective at extracting spatial features from RGB-D data. However, such methods often struggle with the complexities of multi-modality and intricate scene structures. One of the primary issues is that CNNs frequently fail to properly align depth and RGB channels, especially when affected by background noise, occlusions, or misalignment caused by sensor shifts. Additionally, CNNs prioritize local or global features without effectively balancing the two. This can be problematic in tasks like scene classification, where understanding the relationships between objects in space and preserving object boundaries are crucial (*Jia et al., 2021*). Research has further indicated that current methods inadequately address challenges related to sensor misalignment, noise distortion, and multi-modal data fusion, especially when preserving the structural integrity of objects within the scene (*He et al., 2022*). Despite their success in capturing long-range dependencies, attention-based models face challenges in efficiently processing high-resolution RGB-D data. When handling dense spatial information, these models often struggle with computational scalability and memory constraints. Sequence models, while effective for temporal data, have shown limitations in capturing spatial hierarchies and maintaining local feature consistency in RGB-D fusion tasks (*Silberman et al., 2012a*, *2012b*; *Zhang, Li & Zhang, 2020*; *Ahmed & Jalal, 2024a*).

Current transformer-based approaches for RGB-D scene understanding face several key challenges. First, their self-attention mechanisms, while powerful for modeling global relationships, often struggle to effectively balance local and global feature representation (*Meena, Kumar & Yadav, 2024*). Second, the fixed-size patch embedding commonly used in Vision Transformers can lead to information loss at object boundaries and struggle with varying scales of objects in scenes (*Li et al., 2023*). Third, existing attention-based models often fail to effectively handle the unique characteristics of depth information, particularly in cases of sensor noise and missing data (*Park et al., 2023*).

To address these limitations, we propose a novel method that builds upon recent advancements in scene understanding. Our key contributions can be summarized as follows:

- We propose an enhanced Vision Transformer architecture incorporating masked pre-training techniques from MAPE-ViT and adaptive patch integration, enabling better feature learning and scene understanding across different scales.
- We introduce a novel multi-modal feature extraction approach that combines wavelet coefficients with maximally stable extremal regions (MSER), significantly improving both local feature detection and global context understanding in scene analysis.
- We develop an efficient classification framework that integrates an extreme learning machine (ELM) with a Grey Wolf Optimizer (GWO) for optimal feature selection, enhancing the model's accuracy and computational efficiency.
- We demonstrate state-of-the-art performance on both SUN RGB-D and NYU v2 datasets through extensive experimentation, validating the effectiveness of our proposed architecture in indoor scene understanding tasks.
- We provide comprehensive ablation studies and analyses that validate the effectiveness of each component in our proposed framework, offering insights into the contribution of each architectural decision.

The rest of the article is structured as follows: The "Related Work" section represents previous work in this field. "Materials and Methods" covers the proposed methodology, including the stages of pre-processing, RGB and depth image fusion, segmentation, feature extraction, feature-level fusion, and classification. "Results" compares the experimental results with conventional object segmentation, detection, and classification methods. In "Conclusions" we conclude with some key insights and future directions.

## RELATED WORK

In recent years, there's been a lot of progress in detecting and classifying objects in RGB images. Various methods are being used by moving from traditional machine learning to advanced deep learning models. For example, convolutional neural networks (CNN), You Only Look Once (YOLO), Faster region-based convolutional neural network (R-CNN), and solid state drive (SSD) are the most widely used and are well-known for object recognition performance. Meanwhile architectures such as ResNet, Inception, and EfficientNet have proven quite effective for classification. Lately, attention-based models like Vision Transformers have become quite popular, offering new ways to understand images. Researchers have also explored ensemble methods and integrated semantic segmentation with object detection, as seen in Mask R-CNN, to enhance scene comprehension (*Gupta et al., 2014*; *Al Mudawi et al., 2024*; *Zhou et al., 2022a*, *2022b*).

From Table 1, it is clear that multi-object detection and scene classification is still a challenging task for RGB-D images. The comprehensive literature review presented above

**Table 1  Literature review for existing scene classification models.**

| State-of-the-art models | Main contributions | Limitations |
|---|---|---|
| *Ikeda & Ikehara (2023)* | In this article for RGB-D images they propose the saliency and edge reverse attention (SERA) technique, which combines saliency and edge feature fusion with reverse attention mechanisms. The researchers also present the multi-scale interactive module (MSIM) to capture global image information across various scales. | Although the model achieved good accuracy, it considered limited static activities such as drinking glass and pouring water. |
| *Xiong, Yuan & Wang (2021)* | This article proposes a framework for RGB-D scene recognition that adaptively selects key local features to address spatial variability in scene images. A differentiable local feature selection (DLFS) module is used for extracting and selecting image features connected to the scene at both theme and object levels, helping in utilizing the relationship between RGB and depth modalities. A variation mutual information maximization loss is proposed to ensure the selection of discriminative features. The DLFS module is also scalable to different feature sizes, improving overall accuracy. | The construction of the multi-modal graph can be computationally expensive, especially for high-resolution inputs, and the separate CNN backbones for RGB and depth may not fully leverage the complementary nature of the modalities during early feature extraction. |
| *Couprie et al. (2014)* | This article centers on using a multiscale convolutional neural network to learn features directly from RGB and depth images for indoor scene segmentation. The network processes the input at multiple scales using a Laplacian pyramid representation and is trained end-to-end to predict pixel-wise semantic labels for indoor scenes. As a post-processing step, superpixels are employed to smooth the network predictions. The authors utilize temporally consistent superpixels for video sequences to improve frame-to-frame consistency in the segmentation results. | The system is only evaluated on a single dataset NYU Depth V2 which is one main concern about the model's generalization on indoor environment. |
| *Zeng et al. (2019)* | This article involves a multi-modal deep neural network and DS evidence theory for RGB-D object recognition. It preprocesses RGB and depth images and train two convolutional neural networks. Using a quadruplet samples-based objective function, it fine-tunes network parameters for multi-modal feature learning. Two sigmoid SVMs provide probability classification results, fused using DS evidence theory to exploit discriminative and correlation information between modalities. | The model is computationally expensive as it uses multiple neural networks. Moreover, the model's effectiveness depends on the quality and quantity of RGB-D data. |
| *Chen et al. (2021)* | The proposed RD3D model for RGB-D salient object detection introduces a novel approach using 3D convolutional neural networks. It performs pre-fusion of RGB and depth modalities in an inflated 3D encoder, followed by in-depth feature fusion in a 3D decoder equipped with rich back-projection paths (RBPP). This progressive fusion strategy ensures effective integration of RGB and depth streams, enhancing detection accuracy. | The model is computationally expensive, using a 3D convolutional network in encoders and decodes. also, it requires high memory usage. The model uses fix temporal dimensions, which can limit model scalability to other tasks |
| *Jin et al. (2021)* | The proposed complementary depth network (CDNet) integrates RGB and depth streams through four stages: employing VGG encoders for feature extraction, estimating informative depth maps to enhance saliency detection, dynamically fusing depth features based on saliency potential, and employing a two-stage cross-modal fusion scheme. CDNet utilizes original depth maps with high saliency for training and enhances them with estimated depth maps for better performance in RGB-D salient object detection (SOD). This approach aims to leverage both high-level depth features for object localization and low-level features for edge details, improving overall SOD accuracy. | CDNet relies on accurate initial depth maps for effective feature fusion, struggling when maps are of low quality, limiting its predictive capability. Its simplistic depth estimation and reliance on existing RGB-D data hinder robustness in real-world scenarios |

| State-of-the-art models | Main contributions | Limitations |
|---|---|---|
| *Rafique et al. (2022)* | The article proposes a novel multi-object detection and scene recognition model that segments and analyzes objects in RGB and depth images using CNNs. Deep CNN, DWT, and DCT features are extracted from segmented objects and fused in parallel. A genetic algorithm optimizes feature selection for neuro-fuzzy-based object detection and recognition. Object-to-object relations are evaluated *via* probability scores, facilitating scene label prediction with a decision tree. | The model's heavy reliance on pre-trained CNN models may limit its ability to generalize to novel object types or scenes not represented in the training data. |
| *Chen, Li & Su (2019)* | The proposed model adopts a stage-wise training approach to bridge the gap between RGB and depth data distributions and address the scarcity of RGB-D training samples. It initially trains separate RGB-induced and depth-induced saliency detection networks, then combines them into a multi-path, multi-scale, multi-modal fusion network (MMCI net). This approach leverages shared architecture and parameter initialization to enhance model robustness and mitigate overfitting risks, utilizing the same training dataset across all stages for comprehensive learning. | The method relies on a stage-wise training process, where the RGB and depth networks are trained separately before being combined. This may not fully leverage the complementary information between modalities during the initial training stages |
| *Zhou et al. (2023)* | The BCINet model addresses the limitation of integrating higher-level contextual information with lower-level features by introducing a bilateral cross-modal interaction module (BCIM) to bilaterally fuse complementary cues from RGB and depth data, as well as a hybrid pyramid dilated convolution module (HPDC) to capture diverse contextual information along both spatial directions. Additionally, a context-guided module (CGM) is proposed to progressively refine segmentation by transmitting higher-level contextual information to lower-level features. | The BCIM and the hybrid pyramid dilated convolution module HPDC introduce additional computational overhead, which may increase the computational complexity of the model. This could potentially impact the model's efficiency and inference speed. |
| *Ma et al. (2024)* | Adjacent-scale multimodal fusion network (ASMFNet) is a specialized semantic segmentation network for remote sensing data that excels in multimodal feature fusion through its adjacent-scale interaction mechanism. The network's design centers around two key components: the hierarchical scale attention (HSA) module, which processes features at different abstraction levels to understand object-context relationships, and the adaptive modality fusion module, which intelligently combines different sensor modalities using pixel-level spatial weights. Through feature concatenation and filtering, the network evaluates modality importance and integrates cross-modal information effectively, while maintaining computational efficiency. | The model's performance and efficiency are directly tied to how well it can extract multiscale features, suggesting that poor feature extraction could significantly impact results. Moreover, the lack of global modeling capabilities, indicating the model might struggle with capturing long-range dependencies or global scene understanding. |
| *Zhou et al. (2022c)* | Multi-task attention network (MTANet) demonstrates significant advantages over other multi-task learning (MTL) models in medical image analysis, particularly through its innovative attention mechanisms. Unlike traditional MTL approaches that often struggle to balance local and global feature extraction, MTANet employs a reverse addition attention module to enhance segmentation precision by effectively fusing global and boundary cues from high-resolution features. | Although the attention mechanisms in MTANet enhance feature extraction, they may still encounter challenges in capturing long-range dependencies effectively, particularly in complex medical images with significant variability in lesion shapes and sizes. |

(Continued)

| Table 1 (continued) | | |
|---|---|---|
| State-of-the-art models | Main contributions | Limitations |
| *Wan et al. (2023)* | MFFENet, or the multiscale feature fusion and enhancement network, is designed for accurately parsing RGB-thermal urban road scenes. It comprises two encoders that extract features from RGB and thermal images, followed by a feature fusion layer that integrates these multi-scale features. This architecture aims to enhance the robustness of scene parsing, particularly in challenging conditions where traditional methods may struggle. | MFFENet may struggle in extreme weather conditions or varying lighting scenarios, affecting the quality of RGB and thermal images. This sensitivity can lead to inaccuracies in scene parsing when the environmental context changes significantly. |
| *Zhou et al. (2022c)* | The frustum-range networks (FRNet) method aims to restore contextual information in range images by leveraging corresponding frustum LiDAR point clouds. It employs a frustum feature encoder to extract per-point features within the frustum region, preserving scene consistency for point-level predictions. A frustum-point fusion module hierarchically updates the per-point features, allowing each point to capture more surrounding context from the frustum features. Lastly, a head fusion module combines features from different levels to make the final semantic prediction. By fusing range image and LiDAR data, FRNet can effectively incorporate contextual information for improved scene understanding. | The 3D-to-2D projection process can introduce corrupted contextual information, negatively impacting the segmentation quality. This loss of context can lead to inaccuracies in identifying and classifying objects within the scene. |
| *Weng et al. (2024)* | The bimodal fusion rectification network (BFRNet) features a dual-branch architecture for end-to-end semantic segmentation. Its channel and spatial fusion rectification (CSFR) module integrates multimodal features across dimensions. The edge fusion refinement (EFR) module enhances edge feature extraction using bimodal interactive attention, reducing edge loss from single modalities. Lastly, the multiscale feature fusion (MSFF) module combines features from CSFR and EFR for robust multiscale outputs, showcasing improved performance in multimodal data utilization. | The number of pixels representing the background often far exceeds those representing foreground objects. This imbalance can lead to difficulties in training the model effectively, as it may become biased towards predicting the background more frequently than the foreground, especially for smaller or less frequent object. |
| *Zhou et al. (2022c)* | Cross-layer interaction and multiscale fusion network (CIMFNet) offers a multimodal fusion module that explores the similarities and differences between features from both modalities, ensuring a more comprehensive fusion process. This method introduces hierarchical feature interactions by acknowledging the limitations of traditional down sampling operations such as pooling and striding, which can improve feature representativeness but often lead to the loss of spatial details and segmentation errors. This strategy effectively mitigates the adverse effects of downsampling by preserving essential spatial information. Furthermore, implementing a two-way interactive pyramid pooling module allows for the extraction of multiscale contextual features, guiding the feature fusion process and resulting in improved accuracy in segmentation tasks. | The model contends with the high dimensionality of remote sensing data, which can include multi-spectral and hyperspectral images. This complexity can complicate feature extraction and increase computational demands, potentially leading to slower processing times and reduced efficiency. |
| *Li et al. (2024)* | This article presents a unified masked image modeling (MIM) framework for learning representations from multimodal and multi-seasonal remote sensing data. It integrates multimodal data using a concat-style strategy and handles seasonal variations *via* a siamese network. A temporal-Multimodal (TM) fusion block enhances feature fusion during pretraining. The model follows a multi-stage pretraining strategy, transitioning from unimodal to multimodal and finally to seasonal-multimodal learning for comprehensive feature extraction. | The model's performance may degrade under extreme weather conditions or when applied to very high-resolution satellite data. While it supports multimodal data integration, incorporating additional sensor types could further enhance its capabilities. |

| State-of-the-art models | Main contributions | Limitations |
| --- | --- | --- |
| *Wu, Hong & Chanussot (2022)* | The article proposes a novel CNN-based framework for multimodal remote sensing data classification, integrating optical, SAR, and LiDAR data through a hybrid early-late fusion strategy to leverage complementary features. It achieves state-of-the-art accuracy on benchmark datasets, demonstrating robustness across diverse environments and sensor conditions. The architecture incorporates resource-efficient designs, such as lightweight subnetworks and data augmentation, enhancing scalability for real-world applications while addressing challenges like limited labeled data and computational constraints. | The model's computational complexity and reliance on large, co-registered datasets limit practicality in resource-constrained settings. Performance may degrade with domain shifts and misaligned modalities, and the fusion strategy might not fully exploit cross-modal correlations. |
| *Hong et al. (2023)* | The article introduces the C2Seg dataset, a multimodal remote sensing benchmark designed for cross-city semantic segmentation, incorporating hyperspectral, multispectral, and SAR data from urban scenes in Berlin-Augsburg (Germany) and Beijing-Wuhan (China). It proposes HighDAN, a high-resolution domain adaptation network, which uses adversarial learning and a high-to-low resolution fusion strategy to address domain gaps between cities while preserving spatial topology. The model also incorporates Dice loss to mitigate class imbalance issues. | The primary limitation of this work is its focus on only two cross-city scenarios, which may limit the dataset's applicability to broader global urban environments. Additionally, the computational complexity of HighDAN could pose challenges for real-time or resource-constrained applications. |

reveals several critical research gaps in existing RGB-D scene understanding approaches. Current methods like CDNet and BCINet struggle with effective RGB-D fusion, particularly in handling sensor misalignment and noise. While models like Multi-task attention network (MTANet) and multi-modal feature fusion network (MFFNet) attempt to address this through attention mechanisms, they still face challenges in preserving structural integrity during fusion, especially at object boundaries. Current methods predominantly use fixed-size processing units (convolutional kernels or transformer patches), limiting their ability to effectively handle varying object scales. While models like saliency and edge reverse attention (SERA) and differentiable local feature selection (DLFS) attempt to address this through multi-scale processing, they still struggle with dynamic scene compositions. Existing architectures often fail to strike an optimal balance between local and global feature representation. While attention-based models excel at capturing global relationships, they struggle with local feature consistency. These gaps highlight the need for a more comprehensive approach that can effectively balance feature quality, and robustness while maintaining strong performance across varying scales and conditions. Our proposed MAPE-ViT framework aims to address these limitations through its novel integration of adaptive patch embedding, wavelet coefficients, and MSER regions.

## MATERIALS AND METHODS

Figure 1 shows the structural diagram of our proposed model for scene understanding by combining RGB and depth data. Initially, RGB and depth images undergo preprocessing to enhance quality and remove noise. The preprocessed images are fused using our Fusion Model, combining complementary information from both modalities for efficient

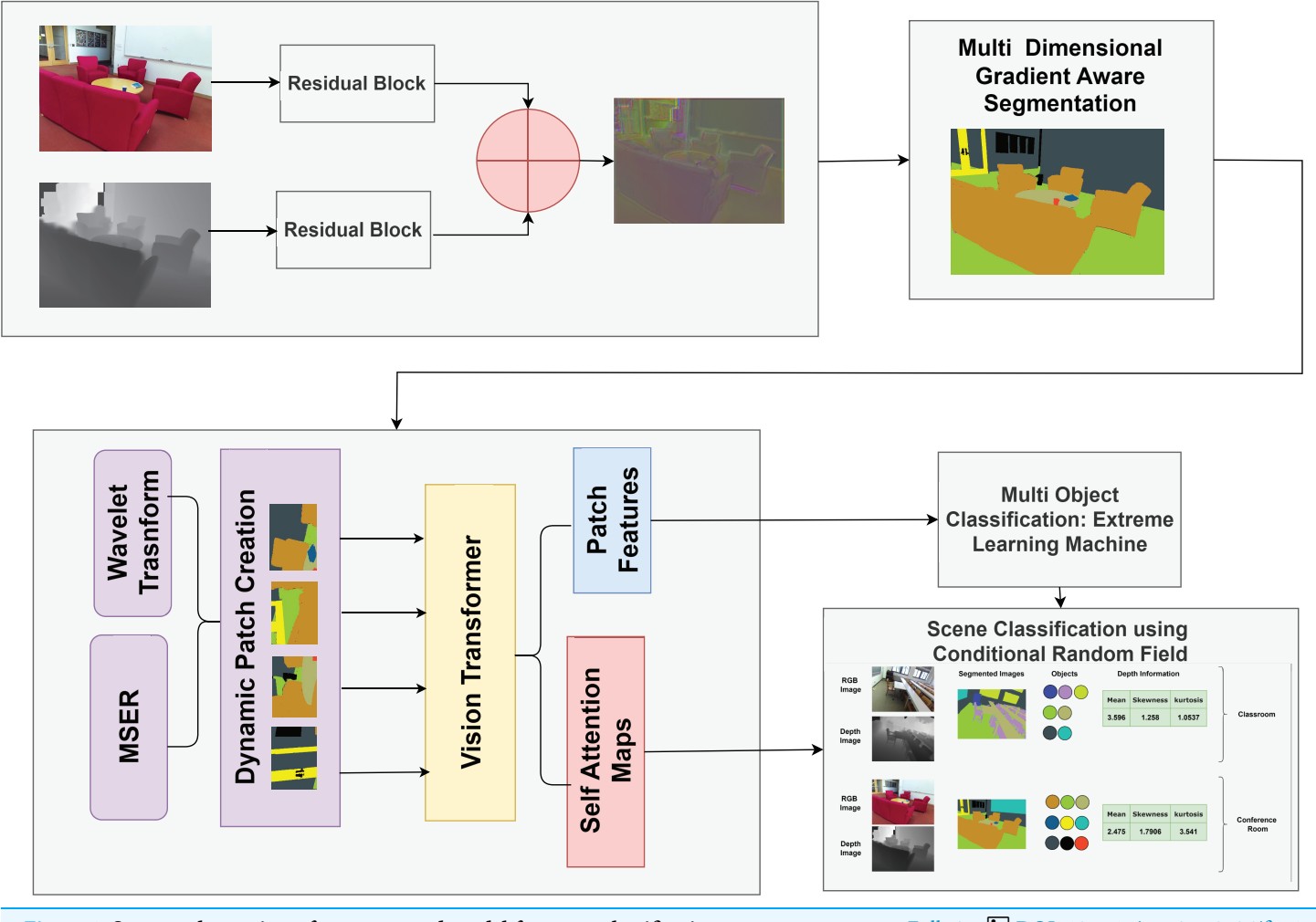

**Figure 1 Structural overview of our proposed model for scene classification.**

processing. The fused images are segmented using the proposed Multi-dimensional Gradient-Aware Segmentation method. We propose a novel feature extraction method, Multimodal Adaptive Patch Embedding with Vision Transformer (MAPE-ViT), which extracts relevant and discriminative features from the segmented images. For multi-object classification, we employ the extreme learning machine (ELM).The ELM classifier takes the optimized features as input and accurately classifies objects in the scene. Lastly, we utilize conditional random fields (CRFs), a powerful probabilistic graphical model for scene classification. The CRF integrates information from multi-object classification results, contextual information extracted using a MAPE-ViT, and depth data from the original images.

## RGBDFusionNet

The fusion of RGB and depth information for indoor scene understanding is challenging due to the variations between these images. The RGB and depth data exhibit different characteristics, making it difficult to fuse them efficiently to prepare them for segmentation

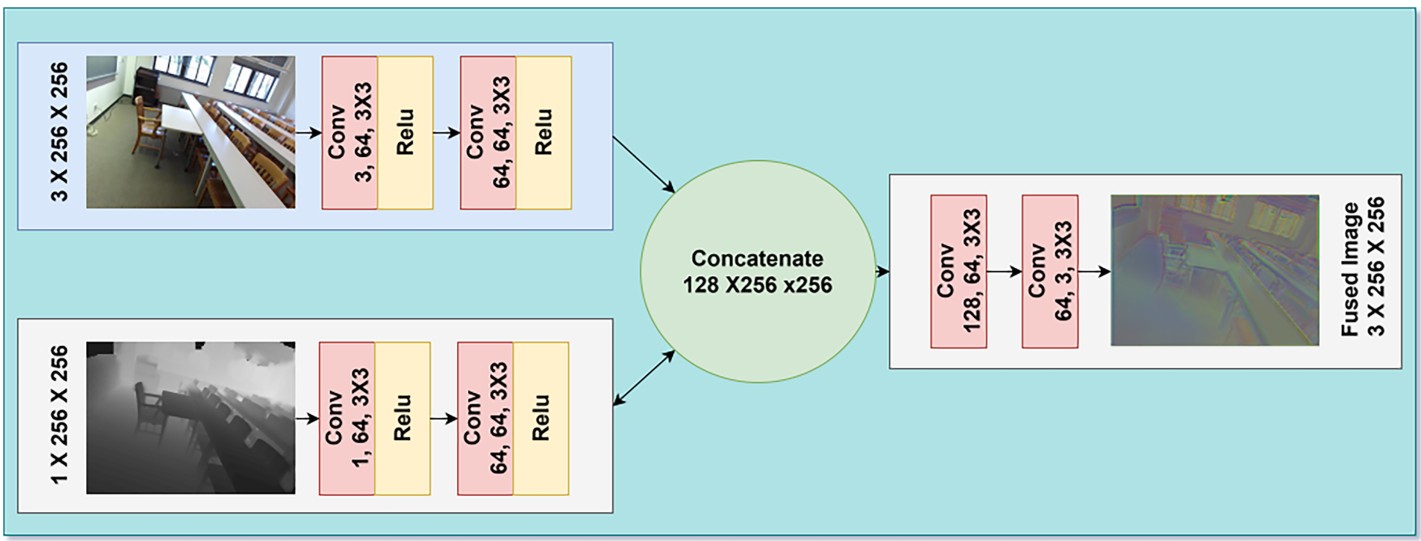

**Figure 2 Architectural diagram of our RGB and depth fusion model.**

tasks (*Farahnakian & Heikkonen, 2022*; *Radford et al., 2021*). To effectively integrate RGB and depth information for indoor scene understanding, we propose the FuseNet architecture, which employs an attention-based fusion strategy. The network architecture is split into two parallel channels in which the first stream handles RGB inputs, while on the other hand the second stream handles the depth data. The RGB and the depth streams are processed in parallel and consist of two residual block and two convolution layers. Each layers has a 3 × 3 kernel which is followed by a batch normalization and ReLU activation function (*Kazakos, Nikou & Kakadiaris, 2018*).

The outputs of these streams are then combined along the channel dimension to create a fused feature map. The important thing to note here is instead of simply element wise adding the feature maps of the two modalities, we incorporate an attention mechanism which helps to weight the features accordingly. This attention module has two convolutional layers. We used 1 × 1 kernel and it will reduce the dimension of the channel to half before passing it to ReLU activation. The second layer which is also 1 × 1 kernels is used to provide a single-channel attention map. A point wise multiplication is performed between attention map and fused feature map in order to emphasize on salient features.

These fused representation from the attention weights are then passed to two more convolutional layers. The first layer consist of 3 × 3 kernel and a ReLU activation function while the second layer consist of 3 × 3 kernel. With the help of this attention-based fusion approach, the proposed FuseNet architecture tries to address issues which arise from the large differences between the RGB and depth modalities, and the nature of depth maps uncertainty. The implementation of selective feature weighting makes it possible for the network to harness complementary information from both modalities; thus, it could enhance performance in indoor scene understanding tasks. The proposed multimodal fusion process is illustrated in Fig. 2 which integrate RGB and depth modalities. The RGB

input undergoes three convolutional layers to extract spatial features, while the depth input is processed through a similar pathway to capture depth-related details. These feature maps are concatenated, forming a unified representation that incorporates both modalities. This fusion enables the model to leverage complementary information from RGB and depth inputs, resulting in more robust image fusion.

## Segmentation

### *Multi-dimensional gradient aware segmentation*

This method is derived from density based clustering that include DBSCAN (*Brahmana, Mohammed & Chairuang, 2020*) with the addition of a distance function tailored for fused image analysis. MGAS uses spatial information, color values together to form local gradients from the fused image and it offers a similarity measure for each pixel.

The prosped distance metric between two pixels *p and q* is defined as

$$d(p,q) = \sqrt{\omega_s \cdot d_{spatial}(p,q)^2 + \omega_c \cdot d_{color}(p,q)^2 + \omega_g \cdot d_{gradient}(p,q)^2} \qquad (1)$$

here, $d_{spatial}(p,q)$ is the euclidean distance in pixel coordinates, $d_{color}(p,q)$ is the difference in color values, $d_{gradient}(p,q)$ is the combined difference in local color gradients, $\omega_s$, $\omega_c$ *and* $\omega_g$ are weights for the spatial, color and gradient components, respectively (*Ahmed & Jalal, 2024b*). Each component of the distance metric is defined as follows:

$$d_{spatial}(p,q) = \sqrt{(P_x - q_x)^2 + (p_y - q_y)^2} \qquad (2)$$

where, $(P_x - p_x)$ and $(q_x - q_y)$ are the cordiantes of pixel p and q.

$$d_{color}(p,q) = \sqrt{(R(p) - R(q))^2 + ((G(p) - G(q))^2 + (B(p) - B(q))^2} \qquad (3)$$

where, $R(p), G(p), B(p)$ and $R(q), G(q), B(q)$ are the red, green and blue color values at pixels p and q, respectively.

$$d_{gradient}(p,q) = \sqrt{(G_{xc}(p) - G_{xc}(q))^2 + (G_{yc}(p) - G_{yc}(q))^2}. \qquad (4)$$

This approach preserves object boundaries by emphasizing local gradients and spatial coherence, addressing the challenge of boundary degradation in cluttered scenes. By integrating depth gradients into the distance metric, it enhances segmentation accuracy in regions with ambiguous color/texture but distinct geometric profiles, directly improving scene parsing for complex indoor environments. Figure 3 illustrates the stages of data processing using our novel fused and segmentation approach; column 1 shows RGB images, which provide the color and texture information of the scenes. Column 2 shows depth images, column 3 depicts fused images that combine the complementary features of RGB and depth modalities, enhancing the richness of extracted features, and column 4 displays the segmented images showcase the final semantic segmentation results, where different objects in the scene are accurately identified and labeled with distinct colors.

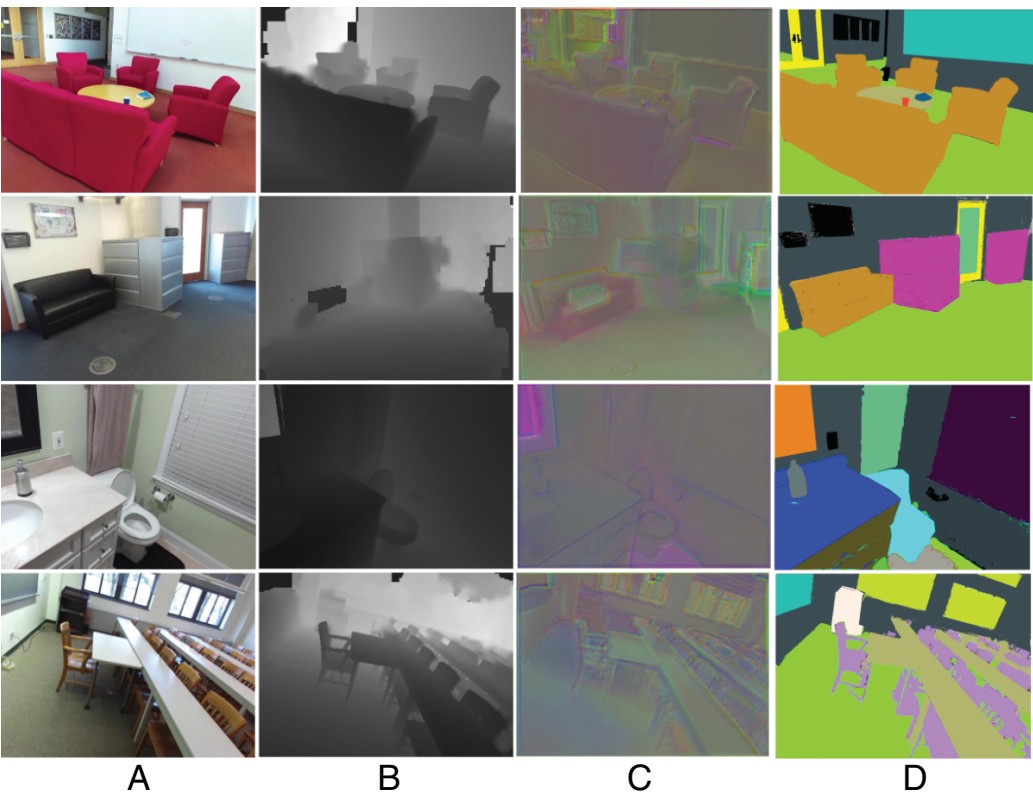

**Figure 3** Segmentation results (A) RGB images (B) depth images (C) fused images and (D) segmented images.

## Feature extraction

This innovative feature extraction method for RGB-D images integrates several advanced techniques in order to create an adaptable approach for image analysis. We propose a novel MAPE-ViT feature extraction which initially, the feature detection involves the identification of maximally stable extremal regions that ensure the stability of regions of interests in an image. The next step is wavelet transform which presents the image at different scales and with its coefficients selectively considering global and local features. The method innovatively combines the information from MSER and wavelet transform, affecting the weighting by MSER regions to the wavelet transform. The final transformation yields a concentration of the enhanced representations of principal image features. This is followed by segmenting the image into adaptive patches where each patch size depends on the MSER region's position. In the following, these patches reserve the original image pixels together with their wavelet coefficients as well as the data that relate to the corresponding MSER region. These multi-modal patches are then re-shaped into an appropriate format to be fed into a Vision Transformer. In this framework, the Vision Transformer is a feature extractor, which conducts the attention mechanism to work with the patch embeddings and generate the features on higher levels. These extracted features can then be used as input to various classifiers for the final image classification task.

### Maximally stable external region

MSER plays an important role in our novel feature extraction model; it identifies stable and distinctive regions within the fused image by detecting blob-like structures that remain consistent across various threshold values, which helps to enhance the overall feature extraction process. The algorithm operates by thresholding the image at all possible intensity levels and identifying connected components exhibiting stability (*Zhao, Wang & You, 2023*). For each threshold t, the stability of a region $Q_t$ is computed using: Eq. (5):

$$q(Q_t) = \frac{\left|\frac{Q_{t+\Delta}}{Q_{t-\Delta}}\right|}{Q_t} \tag{5}$$

where $\Delta$ represents a small change in threshold, and $|\cdot|$ denotes the cardinality of a set. A region is maximally stable at $t*$ if:

$$q(t^*) < q(t^* \pm \Delta) \; for \; some \; small \; \Delta > 0. \tag{6}$$

The MSER outputs a set of stable regions {RD1, RD2, RD3,……RDn} for the depth component. Each Ri is associated with a stability score which can be calculated using Eq. (7):

$$S_i = \frac{1}{q(t_i)} \tag{7}$$

where $t_i$ is the threshold at which Ri is maximally stable. The use of MSER in this feature extraction method offers several advantages. MSER regions are invariant to affine intensity changes, ensuring robustness to lighting variations and minor viewpoint changes, which is valuable for real-world image classification. MSER excels at detecting regions distinctly different from their surroundings, often corresponding to important object parts or textures, aiding in identifying salient, discriminative features (*Tripathi & Rani, 2024*; *Matas et al., 2004*).

### Wavelet transform

Following the MSER computation, we apply a 3D discrete wavelet transform (3D-DWT) to the entire segmented image. The main idea of selecting of 3D-DWT for integration with Vision Transformers is motivated by several key advantages as it provides multi-resolution analysis through one approximation and seven detail coefficient subbands that naturally complement the patch-based architecture of Vision Transformers, it also captures directional information through detail coefficients (cH, cV, cD) and cross-detail coefficients (cHV, cHD, cVD, cHVD) that enhance spatial relationship understanding, moreover it efficiently integrates with MSER regions through our weighting scheme to emphasize stable features, and lastly, it maintains computational efficiency while providing rich feature representations compared to alternatives like Fourier transforms or Gabor filters. Combining wavelets with Vision Transformers thus creates a powerful framework for extracting comprehensive features from RGB-D data. The 3D-DWT decomposes the input depth image I(x,y,z) into one approximation coefficient subband cA and seven detail

coefficient subbands cH, cV, cD, cHV, cHD, cVD, cHVD at each decomposition level (*Al-Qerem et al., 2020*).

### Combination of MSER and wavelet

In this step, we leverage the MSER regions identified in step 1 to guide the emphasis of wavelet coefficients obtained in step 2. This combination allows us to focus on the most stable and informative parts of the image in both color and depth domains.

For each wavelet coefficient w, we compute a weight $\alpha(\omega)$ based on its overlap with the MSER regions as shown in Eq. (8):

$$\alpha(\omega) = 1 + \beta \sum_{i=1}^{M} (s_i \cdot overlap(\omega, R_i)) \tag{8}$$

where $M$ is the total number of MSER regions from RGB and depth components. $s_i$ is the stability score of the i$^{th}$ MSER region, $overlap(\omega, R_i)$ is the degree of overlap between the wavelet coefficient w and the i-th MSER region, $\beta$ is a scaling factor to control the influence of MSER regions.

We then obtain the emphasized wavelet coefficients W by:

$$W' = \{\alpha(\omega) \cdot \omega | \omega \in W\}. \tag{9}$$

This process enhances the wavelet coefficients corresponding to stable regions in both color and depth data, effectively combining the strengths of MSER and wavelet analysis.

### Patch creation

Using the information from the previous steps, we create adaptive patches that focus on the most informative areas of the RGB-D image. The size of each patch P(x,y,z) is determined by Eq. (10).

$$P(x, y, z) = Pmin + (Pmax - Pmin) . \max(S_i . R_i(x, y, z)) \tag{10}$$

where *Pmin* and *Pmax* are predefined minimum and maximum patch sizes, $S_i$ is the stability score of the ith MSER region and $R_i(x, y, z)$ is the indicator function of the ith MSER region. This adaptive approach ensures that larger patches are created around more stable regions, allowing for more detailed feature extraction in these areas (*Dosovitskiy et al., 2020*). Each patch contains original RGB-D image voxels $I_p$ emphasized 3D wavelet coefficient $W_P$ and MSER information $M_P$.

### Patch embedding and vision transformer input

In the final step, we create embedding for each patch and prepare them as input for a Vision Transformer. The patch embedding $e_p$ is computed as

$$e_p = \left[ f_{RGB-D}(I_p); f_{wavelet}(W_p); f_{MSER}(M_p); f_{Spatial}(x_c, y_c, Z_c) \right] \tag{11}$$

where $f_{RGB-D}$ is a function that processes the original RGB-D voxels, $f_{wavelet}$ processes the emphasized wavelet coefficients, $f_{MSER}$ incorporates the MSER information, and $f_{Spatial}$ encodes the spatial position of the patch center $(x_c, y_c, Z_c)$. These embeddings are then normalized and arranged into a sequence.
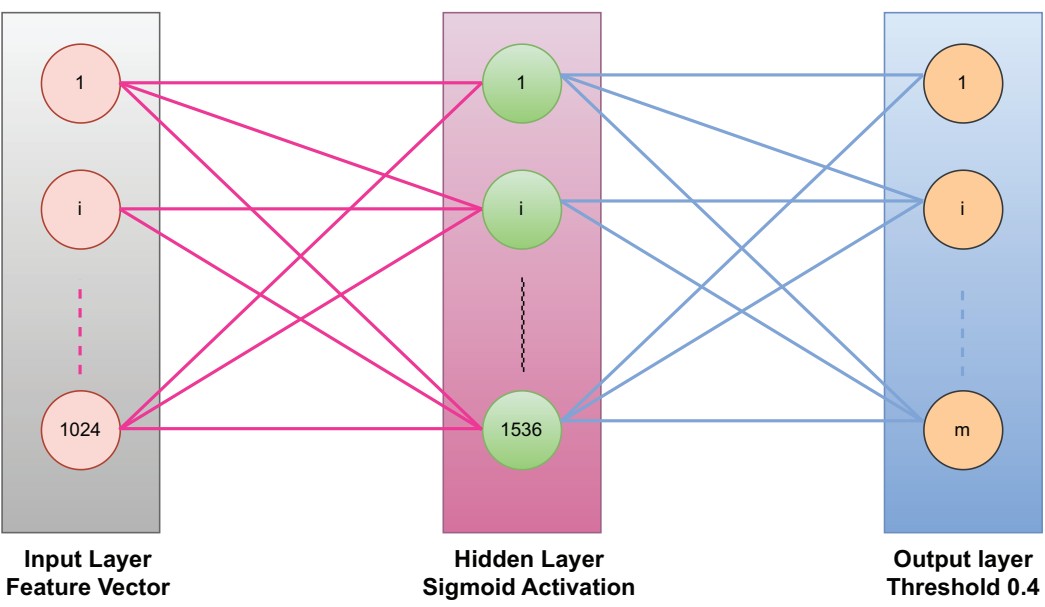

**Figure 4 Extreme learning machine used in our model for classification.**

$$E = [e1_{norm}, \ e2_{norm}, \ \ldots\ldots, \ eN_{norm}]. \tag{12}$$

Finally, we prepare the information for Vision Transformer for feature extraction.

$$Z_0 = [e_{class}; E] + E_{pos} \tag{13}$$

where, $e_{class}$ is a learnable classification token and $E_{pos}$ is positional encoding, which are finally fed to transformer encoder for feature extraction process.

## Extreme learning machine for multi-object recognition

The extracted features sets obtained from MAPE-ViT are fed into an ELM for multi-object recognition. The ELM architecture is chosen for its rapid training speed and strong generalization capabilities which makes it suitable for complex multi-object classification tasks with varying numbers of objects per image (*Wang & Wang, 2021*).

### ELM architecture

The ELM consist of three layers, the first layer is the input layer, then a single hidden layer, and third layer is the output layer. The input layer contains 128 nodes which corresponds to the dimension of the optimized feature vectors. The hidden layer has 512 neurons, which are determined through extensive experimentation in order to balance performance and computational efficiency. The last layer which is the output layer contains m nodes, where m represents the total number of possible object classes in the dataset. Figure 4 represents an ELM with three layers: an input layer (1,024 neurons) for receiving feature vectors, a hidden layer (1,536 neurons) with a sigmoid activation function for nonlinear transformations, and an output layer for generating

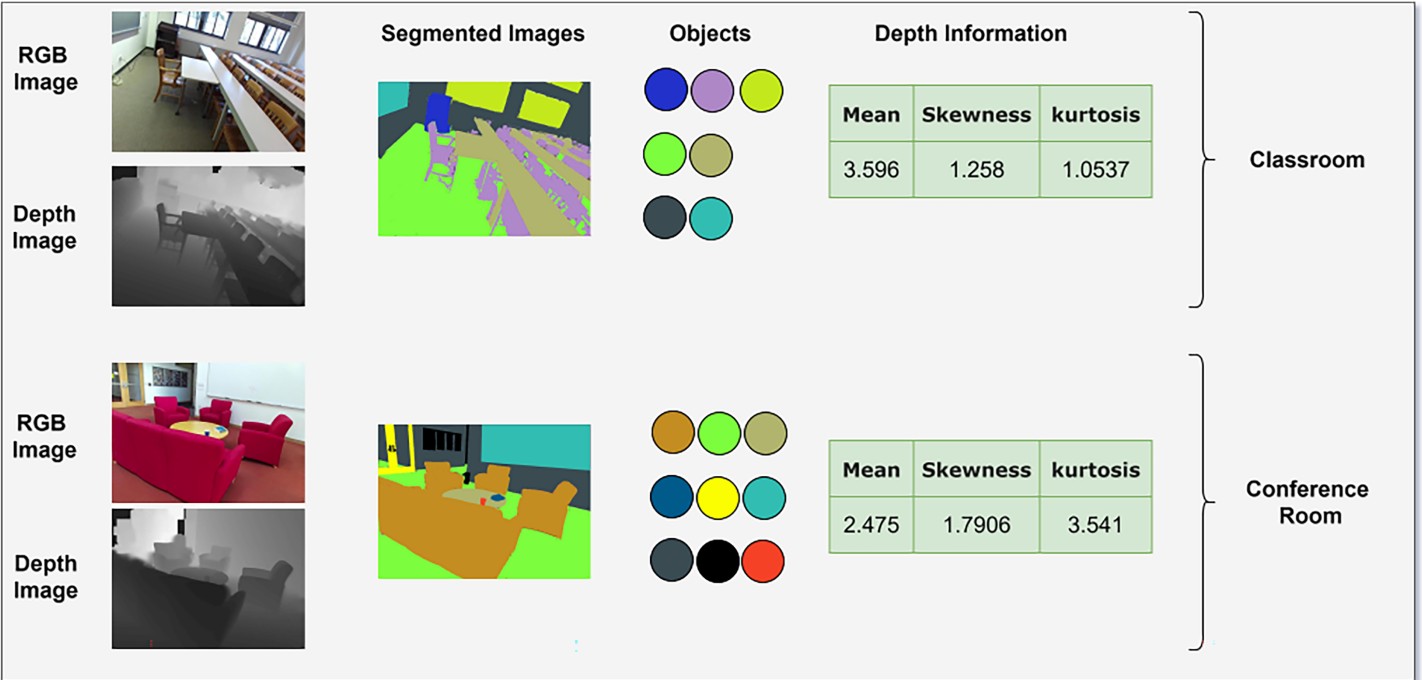

**Figure 5 Scene classification using CRF taking input as multi objects, attention maps and depth information.**

predictions. The input and hidden layers are fully connected with fixed random weights, while the hidden-to-output connections are trained. The output layer applies a threshold of 0.4 for binary classification. This structure ensures fast training and efficient computation.

The ELM output is interpreted as a set of probability scores for multi-object recognition for each object class. The final output is computed as:

$$output = H\beta. \tag{14}$$

A threshold of 0.4 is applied to the output probabilities to determine the presence of each object class. This thresholding approach allows for recognizing a variable number of objects in each image.

## Scene classification with conditional random field

For scene classification, we used a conditional random field (CRF) which will receive three inputs: multi-object classification probabilities, contextual information from the Vision Transformer output, and depth gradient statistics of the images. Depth gradient statistics describe spatial and geometric properties of depth images which are essential for describing scene structure. These gradients are then analyzed to extract various statistical measures that describe the distribution and behavior of the depth gradients across the image. Specifically, we compute mean, variance, skewness, and kurtosis.

### CRF formulation for scene classification

The CRF (*Cao et al., 2020*) is employed to refine scene classification by integrating multi-object classification probabilities, contextual information from the Vision Transformer, and depth gradient statistics. The CRF models the conditional probability of scene labels $Y$ given the input features $X$ as follows:

$$P(Y \mid X) = \frac{1}{z(X)} exp\left( \sum_i \psi_u(Y_i, X) + \sum_{i,j} \psi_p(Y_i, Y_j, X) \right) \quad (15)$$

where $Z(X)$ is the partition function ensuring normalization, $\psi u$ is the near potential, which captures the relationship between each scene label and the input features. It is computed using:

$$\psi_u(Y_i, X) = \omega_{MO} . P_{MO}(Y_i) + \omega_{DGS} . DGS(Y_i) \quad (16)$$

where, $\omega_{MO}$ *and* $\omega_{DGS}$ are the weights learned during the training. $P_{MO}(Y_i)$ is multi object classification probabilities, and $DGS(Yi)$ are the depth gradient statistics (*Li et al., 2020*). Similarly, the pairwise potential $\psi_p$ models the dependencies between neighborhood scene labels as shown in Eq. (17)

$$\psi_p(Y_i, Y_j, X) = \omega_{pair} . exp\left( -||X_i - X_j||^2 \right) \quad (17)$$

where $\omega_{pair}$ is a weight that controls the strength of the pairwise relationship, and $X_i$ and $X_j$ are the input features at positions $i$ and $j$, respectively. Figure 5 illustrates the classification of the scene using CRF that combines probability of multiple objects predication by ELM, contextual information of Vision Transformer outputs and DGS that quantifies the geometrical features using statistical measure as mean, skewness and Kurtosis. The CRF leverages these features to refine scene labels and ensure spatial coherence.

## COMPUTING INFRASTRUCTURE

We conducted the experiments using a PC which is equipped with an x64-based Windows 10 operating system, an Intel Core i3-4010U 1.70.GHz CPU, 4GB RAM. The system's performance was evaluated using two benchmark datasets: SUN RGB-D and NYU v2. Table 2 represents the hyper parameter of MAPE-ViT framework.

## DATASETS DESCRIPTION

The NUY-Dv2 (*Silberman et al., 2012a*) dataset contains labeled and unlabeled images from indoor scenes. The dataset consists of different scenes which includes bathroom, bedroom, bookstore, cafe, kitchen, living room, and office. These scenes encompass a diverse range of objects, including beds, bookshelves, books, cabinets, ceilings, floors, pictures, sofas, tables, TVs, walls, windows, and various background elements. We also used Sun-RGBD (*Silberman et al., 2012a*) dataset for the experiment purposes which also include scenes from indoor environment. From both the datasets we chose 10 common classes for our experimentation.

**Table 2 Hyper parameters for MAPE-ViT framework.**

| Component | Task | Hyperparameter | Value |
|---|---|---|---|
| FuseNet (RGB-D fusion) | Modality fusion | Attention layers | 2 |
| | | Kernel size (RGB/Depth streams) | $3 \times 3$ |
| | | Batch size | 32 |
| | | Learning rate | 0.0001 |
| MGAS (Segmentation) | Scene segmentation | $\in$ (DBSCAN threshold) | 0.5 |
| | | Spatial weight ($ws$) | 0.4 |
| | | Color weight ($wc$) | 0.3 |
| | | Gradient weight ($wg$) | 0.3 |
| MAPE-ViT (Feature extraction) | Feature extraction | Adaptive patch size range | $8 \times 8$ to $32 \times 32$ |
| | | Wavelet decomposition levels | 3 |
| | | MSER stability threshold ($\Delta$) | 0.1 |
| | | Wavelet coefficient scaling ($\beta$) | 0.8 |
| | | ViT patch size | $16 \times 16$ |
| | | ViT heads (multi-head attention) | 8 |
| | | ViT layers | 12 |
| | | Embedding dimension | 768 |
| | | Learning rate | 0.00005 |
| CRF | Scene classification | CRF learning rate | 0.001 |
| | | CRF iterations (mean-field) | 10 |
| | | $\omega_{MO}$ (object weight) | 0.6 |
| | | $\omega_{DGS}$ (depth weight) | 0.4 |

**Table 3 Confusion matrix result for scene classification over NYU-Dv2 dataset.**

| Classes | BDR | KIT | LVR | BTR | DRM | OFF | HOF | CLS | BST | OTH |
|---|---|---|---|---|---|---|---|---|---|---|
| BDR | 0.802 | 0 | 0.181 | 0 | 0.017 | 0 | 0 | 0 | 0 | 0 |
| KIT | 0 | 0.902 | 0.048 | 0 | 0.050 | 0 | 0 | 0 | 0 | 0 |
| LVR | 0 | 0 | 0.845 | 0 | 0.155 | 0 | 0 | 0 | 0 | 0 |
| BTR | 0 | 0 | 0 | 0.923 | 0 | 0 | 0 | 0 | 0 | 0.077 |
| DRM | 0 | 0 | 0.023 | 0 | 0.987 | 0 | 0 | 0 | 0 | 0 |
| OFF | 0 | 0 | 0 | 0 | 0 | 0.762 | 0.238 | 0 | 0 | 0 |
| HOF | 0 | 0 | 0 | 0 | 0 | 0.206 | 0.804 | 0 | 0 | 0 |
| CLS | 0 | 0 | 0 | 0 | 0.115 | 0 | 0 | 0.885 | 0 | 0 |
| BST | 0 | 0 | 0 | 0 | 0 | 0 | 0 | 0 | 0.983 | 0.017 |
| OTH | 0 | 0 | 0 | 0 | 0 | 0.015 | 0 | 0 | 0 | 0.985 |
| Mean ACC = 88.79% | | | | | | | | | | |

Note:
BDR, Bedroom; KIT, kitchen; LVR, living room; BTR, bathroom; DRM, dining room; OFF, office; HOF, house office; CLS, Classroom; BST, book store; OTH, others.

Table 4 Confusion matrix result for scene classification over SUN RGB-D dataset.

| Classes | BDR | KIT | LVR | BTR | DRM | OFF | HOF | CLS | BST | OTH |
|---|---|---|---|---|---|---|---|---|---|---|
| BDR | 0.785 | 0 | 0.145 | 0 | 0 | 0 | 0 | 0 | 0 | 0.070 |
| KIT | 0 | 0.811 | 0 | 0.029 | 0 | 0 | 0 | 0 | 0.039 | 0.121 |
| LVR | 0.197 | 0 | 0.722 | 0.034 | 0 | 0 | 0 | 0.047 | 0 | 0 |
| BTR | 0 | 0.038 | 0 | 0.783 | 0 | 0.048 | 0 | 0 | 0 | 0.131 |
| DRM | 0 | 0 | 0.083 | 0 | 0.837 | 0 | 0.020 | 0.060 | 0 | 0 |
| OFF | 0 | 0 | 0 | 0 | 0.019 | 0.795 | 0.137 | 0.049 | 0 | 0 |
| HOF | 0 | 0.159 | 0 | 0 | 0.054 | 0.227 | 0.672 | 0 | 0 | 0.112 |
| CLS | 0 | 0.029 | 0 | 0 | 0 | 0.198 | 0 | 0.772 | 0 | 0.001 |
| LIB | 0 | 0 | 0 | 0 | 0 | 0.047 | 0.054 | 0 | 0.881 | 0.018 |
| OTH | 0.073 | 0.015 | 0 | 0.036 | 0 | 0.058 | 0 | 0.020 | 0 | 0.798 |

Mean ACC = 78.56%

Note:
   BDR, Bedroom; KIT, kitchen; LVR, living room; BTR, bathroom; DRM, dining room; OFF, office; HOF, house office; CLS, Classroom; LIB, Library; OTH, others.

Table 5 Precision, recall, and mAP results over NYU-Dv2 dataset.

| Classes | Precision | Recall | mAP |
|---|---|---|---|
| BDR | 1.000 | 0.962 | 0.962 |
| KIT | 1.000 | 0.979 | 0.979 |
| LVR | 0.815 | 0.845 | 0.689 |
| BTR | 1.000 | 0.953 | 0.953 |
| DRM | 0.844 | 0.977 | 0.824 |
| OFF | 0.754 | 0.782 | 0.590 |
| HOF | 1.000 | 0.904 | 0.904 |
| CLS | 1.000 | 0.895 | 0.895 |
| BST | 1.000 | 0.973 | 0.973 |
| OTH | 0.905 | 0.935 | 0.846 |
| Mean | 0.931 | 0.920 | 0.862 |

# RESULTS

The experimental results presented in the article demonstrate the effectiveness of the proposed method in scene classification tasks using RGB-D data. The combination of MSER, wavelet transforms, and Vision Transformers for feature extraction has proven to be a robust and discriminative approach, overcoming challenges related to sensor misalignment, depth noise, and object boundary preservation. One of the key strengths of our method is its ability to accurately classify scenes with distinct visual characteristics and depth patterns. It was highly accurate with offices, bedrooms, and living rooms, primarily due to clearer structural patterns and furniture arrangements. This can be attributed to the introduction of MSER-guided image patches, and Vision Transformers whereby these methodologies are capable of independent scene analysis due to their inherent capability of

**Table 6 Precision, recall, and mAP results over SUN-RGBD dataset.**

| Classes | Precision | Recall | mAP |
|---|---|---|---|
| BDR | 0.724 | 0.685 | 0.496 |
| KIT | 0.802 | 0.811 | 0.650 |
| LVR | 0.780 | 0.722 | 0.563 |
| BTR | 0.765 | 0.783 | 0.599 |
| DRM | 0.813 | 0.837 | 0.680 |
| OFF | 0.671 | 0.695 | 0.466 |
| HOF | 0.728 | 0.672 | 0.489 |
| CLS | 0.772 | 0.772 | 0.596 |
| BST | 0.854 | 0.881 | 0.752 |
| OTH | 0.815 | 0.798 | 0.650 |
| Mean | 0.772 | 0.765 | 0.594 |

capturing both local and global features, object boundaries, as well as spatial relationships. The classification results of both datasets are shown in Tables 3 and 4 using confusion matrix which is one of the ways to display the detailed breakdown of the model's performance for all the classes in dataset. It highlights the insight of the scene or object categories which confuses with one another. It displays true positives (TP), false positives (FP), false negatives (FN), and true negatives (TN) for each class present in the dataset.

The confusion matrices reveal that bedrooms in both datasets suffer from FP with living rooms due to shared structural features, such as beds and sofas. Similarly, kitchens and dining rooms exhibit mutual confusion, primarily caused by overlapping objects like countertops and tables. In the NYU-Dv2 dataset, living rooms are often misclassified as dining rooms (15.5% FP rate) due to shared furniture layouts. On the SUN RGB-D dataset, bathrooms face challenges with other categories (13.1% FP rate), potentially due to clutter or poor lighting. Lastly, offices and house offices show mutual misclassification across both datasets, reflecting difficulties in distinguishing desk-based environments with similar setups.

The performance of a model is often evaluated using key metrics: precision, recall and mAP as displayer in Tables 5 and 6. Precision measures the accuracy of the positive predictions made by the model. It indicates the proportion of true positive instances among all instances that the model identified as positive. This means that a high precision score reflects a model that returns mostly relevant results, minimizing the number of incorrect positive predictions. Conversely, recall assesses the model's ability to identify all relevant instances within a dataset. It captures how many true positive instances were correctly predicted by the model out of all actual positive instances. The mAP incorporates the trade-off between precision and recall and considers both false positives (FP) and false negatives (FN).

$$\text{Precison} = \frac{\text{TP}}{\text{TP} + \text{FP}} \qquad (18)$$

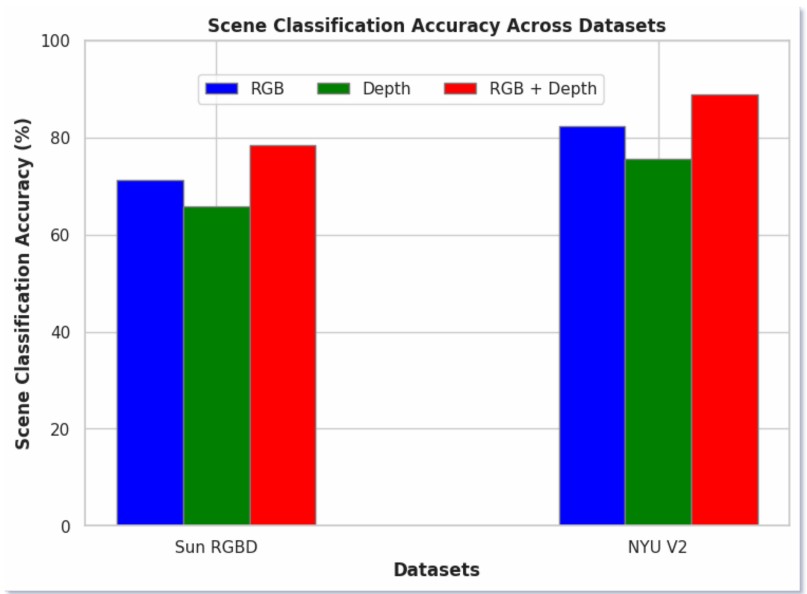

**Figure 6 Scene classification using RGB, depth and both RGB and depth images.**

**Table 7 Comparison of proposed method with existing approaches.**

| SOTA methods | SUN RGB-D | NYUv2 |
|---|---|---|
| *Song, Chen & Jiang (2017)* | – | 66.9 |
| *Song et al. (2018)* | 53.8 | 67.5 |
| *Xiong, Yuan & Wang (2020)* | 56.2 | 68.1 |
| *Du et al. (2019)* | 56.7 | 69.2 |
| *Rafique et al. (2022)* | 63.1 | 72.8 |
| *Cai & Shao (2019)* | 48.7 | 79.3 |
| *Seichter et al. (2022)* | 61.8 | 76.5 |
| *Pereira et al. (2024a)* | 62.3 | 77.8 |
| *Pereira et al. (2024b)* | 63.7 | 80.1 |
| Proposed model | 78.56 | 88.79 |

$$Recall = \frac{TP}{TP + FN} \qquad (19)$$

$$mAP = \frac{1}{N}\sum_{i=1}^{N} AP_i. \qquad (20)$$

Our proposed model performs remarkably in both scene classification and object segmentation tasks on the challenging NYU-Dv2 and SUN-RGBD datasets. Notably, the model identifies specific scene types like bedrooms, kitchens, bathrooms, closets, and bookstores, exhibiting near-perfect precision and high recall. However, it encounters some challenges with visually similar or cluttered scenes such as living rooms, dining rooms,

offices, and home offices, particularly on the NYU-Dv2 dataset, where either precision or recall suffers. Interestingly, the model performs relatively better across all scene categories on the SUN-RGBD dataset, potentially due to cleaner or more distinct scene representations.

### Ablation study

In Fig. 6, we present the ablation study of our proposed model, comparing the scene classification performance using RGB images, depth images, and a fused approach. The results demonstrate that the fused images consistently yielded the highest classification accuracy across both datasets. Specifically, RGB images achieved 72% accuracy on the SUN RGB-D dataset and 82% on the NYU v2 dataset. Depth images, on the other hand, produced 67% and 76% accuracy on the respective datasets. In contrast, the fused RGB-D images significantly outperformed the individual modalities, achieving 78.56% accuracy on the SUN RGB-D dataset and 88.79% on the NYU v2 dataset. The better performance of the fused approach highlights the complementary nature of RGB and depth information and when combined, it provides a better understanding of the scene. This fusion leverages the strengths of both modalities, capturing both the color and texture information from RGB images and the spatial and structural details from depth images.

Table 7, shows the comparison of our proposed model with the state of the art model. Our model outperformed SOTA methods in both of the datasets by achieving the 78.56% and 88.79% scene classification accuracies on SUN RGB-D and NYUv2 datasets respectively. These results are credited to our novel feature extractor methods which extract features for classification and also as we provide very enrich input to CRF model for scene classification which comprises of multi-object classification probabilities, contextual information from the Vision Transformer, and depth gradient statistics.

## DISCUSSION

The experiments conducted on our novel MAPE-ViT model for multimodal scene understanding have yielded promising results, validating the effectiveness of our approach in addressing key challenges in RGB-D image classification specifically issues related to sensor misalignment, noise distortion, and the fusion of multi-modal data. It demonstrates exceptional performance across various environmental conditions and complex scene structures. The MAPE-ViT model, with its innovative combination of MSER and wavelet coefficients, extracts important information from complex multimodal data. By integrating RGB and depth information through our adaptive patch embedding technique, it captures both local and global features which is one of the fundamental factor for accurate scene classification. The incorporation of Vision Transformers for processing these patch embedding further enhances the model's ability to handle intricate spatial relationships and sensor misalignment issues which is one of the common challenge in RGB-D fusion. We used the Grey Wolf Optimizer for feature selection and the extreme learning machine for final classification which is shown significant improvements in both accuracy and efficiency compared to traditional approaches. As a result it helps to take informed

decision for applications like autonomous navigation and augmented reality as it enhances the object detection and classification process.

## LIMITATIONS

Although our model has produced good results across both the datasets. They key aspects for achieving state of the art results across both dataset is the multi modal data fusion and novel MAPE-ViT based feature extraction. However, there are some challenges and limitations like our model is evaluated on two datasets SUN RGB-D and NYU v2, both these datasets focus mainly on the indoor scenes. So, this limits our model generalizability on outdoor scenes which will be addressed in future work. Another important thing to note is that the attention method used in MAPE-ViT, heavily relies on global self attention. This make it less focused on local details as a result the objects with larger size and dominate features plays vital role in feature extractions which some times can add bias towards larger objects in scenes. The proposed framework is specifically designed and tested for RGB-D images. It does not currently support data from other sensors, such as LiDAR or thermal imaging. Adapting the model to handle such modalities would require additional research and modifications to the data fusion and feature extraction mechanisms, which we plan to explore in subsequent studies. The attention method in MAPE-ViT relies heavily on global self-attention, which can sometimes overshadow local details. This may introduce bias toward larger or more dominant features in the scene, potentially underrepresenting smaller or subtler objects. We aim to refine the attention mechanism by incorporating localized attention strategies or hierarchical attention layers in future iterations to mitigate this issue.

## CONCLUSION AND FUTURE WORK

In this article, we presented a novel scene classification method that uses novel Multimodal Adaptive Patch Embedding with Vision Transformer for feature extraction which combines MSER, wavelet transforms, and Vision Transformers to effectively address challenges in RGB-D image analysis such as sensor misalignment, depth noise, and object boundary preservation. In our method, we use the MSER stable regions connected with the wavelets coefficients to construct more comprehensive descriptors. The generation of rich, multi-modal patch embedding and their processing by Vision Transformers allow our method to learn complex relationships between RGB and depth information, enabling dual outputs for multi-object and scene classification. In addition, the anisotropic diffusion and Gray Wolf Optimization improve the stability and identification capability of the proposed method as shown in experimental results.

Our research opens up several promising directions for future work, particularly in extending our model's capabilities to diverse domains. A primary direction is adapting the MAPE-ViT framework for outdoor scene understanding, where challenges include varying lighting conditions, weather effects, and dynamic object interactions. This would require enhancing our RGB-D fusion technique to handle larger depth ranges and variable environmental conditions. Another significant direction is extending our framework to remote sensing applications. This would involve adapting MAPE-ViT to

process multi-spectral and hyperspectral imagery alongside RGB-D data. The model's wavelet-based feature extraction and adaptive patch embedding could be particularly valuable for analyzing satellite and aerial imagery, enabling applications such as land use classification, urban planning, and environmental monitoring. We plan to modify our architecture to handle the unique characteristics of remote sensing data, including different spatial resolutions, multiple spectral bands, and varying viewing angles.

### Funding
This work was funded through the Princess Nourah bint Abdulrahman University Researchers Supporting Project number (PNURSP2025R508), Princess Nourah bint Abdulrahman University, Riyadh, Saudi Arabia. The funders had no role in study design, data collection and analysis, decision to publish, or preparation of the manuscript.

### Grant Disclosures
The following grant information was disclosed by the authors:
Princess Nourah bint Abdulrahman University Researchers Supporting Project: PNURSP2025R508.
Princess Nourah bint Abdulrahman University, Riyadh, Saudi Arabia.

### Competing Interests
The authors declare that they have no competing interests.

### Author Contributions
- Muhammad Waqas Ahmed performed the experiments, performed the computation work, prepared figures and/or tables, authored or reviewed drafts of the article, and approved the final draft.
- Touseef Sadiq analyzed the data, performed the computation work, authored or reviewed drafts of the article, and approved the final draft.
- Hameedur Rahman conceived and designed the experiments, performed the experiments, authored or reviewed drafts of the article, and approved the final draft.
- Sulaiman Abdullah Alateyah conceived and designed the experiments, authored or reviewed drafts of the article, and approved the final draft.
- Mohammed Alnusayri performed the experiments, analyzed the data, authored or reviewed drafts of the article, and approved the final draft.
- Mohammed Alatiyyah conceived and designed the experiments, analyzed the data, prepared figures and/or tables, authored or reviewed drafts of the article, and approved the final draft.
- Dina Abdulaziz AlHammadi conceived and designed the experiments, performed the computation work, authored or reviewed drafts of the article, and approved the final draft.

## Data Availability

The code is available in the Supplemental File.

The SUN RGB-D dataset is available at https://rgbd.cs.princeton.edu.

The NUY V2 dataset is available at https://cs.nyu.edu/~fergus/datasets/nyu_depth_v2.html.

## Supplemental Information

Supplemental information for this article can be found online at http://dx.doi.org/10.7717/peerj-cs.2796#supplemental-information.

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
