# Peer review of "MAPE-ViT: multimodal scene understanding with novel wavelet-augmented Vision Transformer"

_PeerJ Computer Science, doi:10.7717/peerj-cs.2796_

## Round 0.1 · original submission · Major Revisions

The reviewers have substantial concerns about this manuscript. The authors should provide point-to-point responses to address all the concerns and provide a revised manuscript with the revised parts being marked in different color.

Reviewer 1 ·

Basic reporting

This paper proposed a novel Wavlet-augmented Vision Transformer for Multimodal scene understanding. Overall, the structure of this paper is well organized. The presentation is clear to the readers. Some detailed comments can be found as follows.
1. It seems that this paper addressed an issue about semantic segmentation in multimodal data. Scene understanding is too big, which still has a distance to achieve the understanding level.
2. Many important works on deep learning for multimodal data are missing and need to be further discuss, such as Convolutional Neural Networks for Multimodal Remote Sensing Data Classification,
3. Please explain why wavlet is a good solution to be selected for Vision transformers ,compared to other transformations or methods.
4. Some visual results can be added to provide a more comprehensive analysis in experiments.
5. How about the future works?

Experimental design

Some visual results can be added to provide a more comprehensive analysis in experiments.

Validity of the findings

Please explain why wavlet is a good solution to be selected for Vision transformers ,compared to other transformations or methods.

Reviewer 2 ·

Basic reporting

The paper focuses on the use of RGB-D data for scene classification and proposes a method that integrates various techniques to address challenges such as sensor misalignment, depth noise, and object boundary preservation, demonstrating certain research value. However, the paper has significant deficiencies in several key areas that require comprehensive and in-depth revisions to meet publishable quality standards in the academic field.
- Insufficient citation and explanation of figures: Although the paper includes multiple figures, the references and explanations for these figures in the main text are inadequate. The meaning and significance of some figures are not well articulated, making it difficult for readers to quickly grasp key information from them. The main text should describe in greater detail the content displayed in the figures and conduct an in-depth analysis based on the results illustrated.
- Language issues: The overall expression of the language in the paper exhibits some unclear and inaccurate elements, with certain statements making it difficult for readers to understand the specific operations and technical details of the model. For instance, the definitions and explanations of some technical terms are not sufficiently clear, and complex sentence structures lead to a lack of logical clarity. It is recommended to polish the language to ensure that the expressions are accurate, concise, and comprehensible.
- Model complexity and interpretability: With the integration of various technologies in the model, the complexity has increased, but the interpretability of its internal mechanisms and decision-making processes remains low. Efforts should be made to employ visualization techniques or other methods to explain how the model processes input data, how feature extraction and classification decisions are made, in order to enhance the model's interpretability.
- Unclear description of methodological innovation: Some techniques proposed in the paper, such as Multi-modal Adaptive Patch Embedding (MAPE-ViT) and Multi-dimensional Gradient-Aware Segmentation (MGAS), although claimed to be novel approaches, do not clearly demonstrate their innovations and advantages when compared to existing technologies. A more explicit explanation should be provided regarding how these methods improve upon existing techniques and address their limitations. The existing relevant works on multimodal semantic segmentation can been discussed, such as BCINet, MTANet, MFFENet, FRNet, PGDNet, CIMFNet.
- In-depth discussion of research limitations: The paper mentions some research limitations, such as those related to the dataset and attention mechanism. However, further exploration of other potential limitations could be beneficial, such as the model's adaptability to different sensor characteristics and computational resource requirements.
- Attention mechanism issues: The attention method used in MAPE-ViT excessively relies on global self-attention, resulting in insufficient focus on local details. During the feature extraction process, large and dominant features may overshadow smaller or less significant objects in the scene, leading to bias in the results.

Experimental design

As above

Validity of the findings

As above

Additional comments

As above

Reviewer 3 ·

Basic reporting

1. There are a lot of grammatical mistakes and improper punctuation throughout the article.
2. The abstract contains many independent sentences without proper continuation "MSER-guided image patches are generated, incorporating original pixels, wavelet coefficients, and MSER information, providing a rich, multi-modal representation.". It also lacks information on metric information.
3. The authors discuss issues related to Convolutional Neural Networks (CNNs) and other conventional paradigms in introduction section but not focused on explaining the challenges in models that use attention mechanisms, sequence models, and other deep learning approaches. Explanation in Introduction section is not adequate.
4. Include organization of the paper at the end of introduction section.
5. The literature review is insufficient to define the problem statement. Include recent papers in the literature review. Min 20+ papers required.
6. Summarize the research gaps at end of Literature review section.

Experimental design

7. Why did the authors not conduct experiments using the "Hypersim" dataset, which is recent dataset? Why did they conduct experiments with old datasets published in 2013?
8. Include more details implementation details about proposed modules such as Pre-processing Technique, ELM Architecture, Multi-dimensional Gradient Aware Segmentation and so on.

Validity of the findings

9. Conduct ablation study by including impact of proposed modules.
10. Analysis the False Positive and False Negative in each class.
12. Most of the existing methods use mean Average Precision (mAP) for multi-class classification. Why did the author choose accuracy and F1-score instead? The mAP evaluation metric equation should be included, along with its results.
13. The proposed model struggles to classify larger objects. What are they?
14. Authors can post the code on GitHub or in the public domain.
15. Authors are requested to list their contributions at the end of the Introduction section.

Additional comments

no comment

---

## Round 0.2 · Major Revisions

The reviewers still think there are some major concerns that haven't been addressed. The authors should provide point-to-point responses to address all the remaining concerns and provide a revised manuscript with the revised parts being marked in different color.

Reviewer 1 ·

Basic reporting

Clarity and Structure: The paper is written in clear, professional English. The introduction provides context, outlining the importance of RGB-D scene classification and challenges like sensor misalignment, depth noise, and object boundary preservation.

Literature Review: The authors provide a comprehensive literature review, citing relevant and recent works. However, a more detailed discussion of the specific gaps in existing models could help strengthen the motivation for the proposed method, such as cross-city matters, SeaMo: A Multi-Seasonal and Multimodal Remote Sensing Foundation Model, Convolutional neural networks for multimodal remote sensing data classification, etc.

Methodology: The methodology section is well-structured, detailing the steps for pre-processing, fusion, segmentation, feature extraction, and classification. However, further elaboration on the specific contributions of each component would benefit the reader.

Experimental design

Datasets: The experiments were conducted on two well-known datasets, SUN RGB-D and NYU v2, which provide a good basis for scene classification research. The use of these datasets is appropriate, although including outdoor scenes in future experiments would broaden the generalizability of the findings.

Experimental Setup: The experimental design includes the use of several advanced techniques (MSER, wavelet coefficients, and Vision Transformers) to extract features, followed by multi-object classification with Extreme Learning Machine (ELM) and scene-level classification using Conditional Random Fields (CRF). However, more details on the computational infrastructure a

Validity of the findings

Results: The results presented are promising, with the proposed method showing significant improvements over existing methods in scene classification accuracy. The fusion of RGB and depth information using adaptive patch embedding and Vision Transformers appears to be a robust solution for handling complex indoor scenes.

Ablation Studies: The authors provide an ablation study that compares different configurations of the model (RGB, depth, and fused images), demonstrating the superiority of the fused approach. This is a key strength of the paper, as it validates the effectiveness of the proposed method.

Limitations: The paper addresses some limitations, such as the model's performance being evaluated only on indoor scenes and the reliance on global self-attention mechanisms, which may overlook local details. These limitations are acknowledged, and future work is proposed to address them, including applying the model to outdoor scenes and refining the attention mechanism.

Additional comments

Novelty and Impact: The proposed method introduces a novel combination of MSER, wavelet coefficients, and Vision Transformers, which is a unique approach for RGB-D scene classification. The method's ability to handle sensor misalignment and depth noise sets it apart from existing models, which is a significant contribution to the field.

Computational Complexity: While the paper does not provide detailed information on the computational complexity or runtime, this aspect should be further explored, especially considering the use of multiple modalities and the integration of complex models like Vision Transformers and CRFs.

Future Directions: The authors propose several promising directions for future work, including extending the model to remote sensing applications, which could open up new avenues for environmental monitoring, urban planning, and other remote sensing tasks.

Reviewer 3 ·

Basic reporting

No comment

Experimental design

No comment

Validity of the findings

No comment

Additional comments

All the comments are well addressed and can be accepted.

---

## Round 0.3 · accepted · Accept

Reviewers are satisfied with the revisions, and I concur to recommend accepting this manuscript.

Reviewer 1 ·

Basic reporting

No more comments.

Experimental design

No more comments.

Validity of the findings

No more comments.

Additional comments

No more comments.